# Are Rainwater and Stormwater Part of the Urban CE Efficiency?

**Carlos Novaes * and Rui Marques** 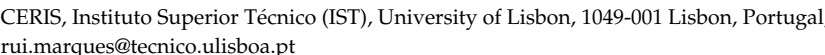

CERIS, Instituto Superior Técnico (IST), University of Lisbon, 1049-001 Lisbon, Portugal;
rui.marques@tecnico.ulisboa.pt
* Correspondence: cnovaes.augusto@gmail.com; Tel.: +55-61-99999-8767

**Abstract:** Circular economy (CE) means efficient resource use. It is a matter of better available resource management. Understanding the characteristics, potential, use advantages and disadvantages, and management systems, in each context, is the basis to construct a feasible CE framework to deal with climate change and economic scarcity challenges. Urban stormwater has potential importance in CE when addressed as a useful resource rather than as waste. Its use can replace part of the water supply (reduce principle), brought from distant sources using energy-consuming and emission-producing systems. Thus, it can be a source of energy savings and emission reduction since stormwater can be used and stored near the place where rainwater falls or infiltrates to supply groundwater (reuse principle). Urban agriculture can also gain benefits by using, e.g., green infrastructures (GIs) (recycling principle). The main gap still lies in the implementation of the efficiency mentality, reducing expenses and consequently improving revenues, profits, and environment issues, such as emissions. It is a big paradigm shift. The creation of policies, institutions, and regulations aligned with each other, together with urban planning and water cycle efficiency, from a CE perspective is fundamental. Urban stormwater as a CE component is a moving paradigm shift based on a change in mindset.

**Keywords:** circular economy; emission reductions; energy savings; green infrastructure; urban stormwater

## 1. Introduction

Stormwater is the portion of rainwater that runs off without evaporating or infiltrating into the ground or, according to the definition by the Environment Protection Authority (EPA) of Australia: "Stormwater is rainwater that flows across outside surfaces into stormwater drains and gutters in the street" (EPA 505/03) [1].

The research question is whether rainwater and stormwater can be included in CE and how they are viewed, or can be viewed, from this perspective, considering their interrelationships with the urban water cycle and the diverse realities in terms of water use efficiency, sanitation, and Sustainable Development Goals (SDGs). The research approach was carried out by consulting the literature involving water and CE.

The main question about rainwater and stormwater in a CE context is about resource use efficiency, and it requires a mindset change driving actions towards sustainable paradigm shift. Conceptually, in terms of the circular economy (CE) model, besides the search for more economic efficiency, according to the idea of a closed flow of materials and energy, there are three Rs: reduce, reuse, and recycle that, for this, must be measured [2]. The evaluation can be comparative, for example, with the alternatives of the linear model or in time-based terms, allowing evaluation of the evolution and improvement of the circular model itself, thus adjusting it. How to measure the circularity of the options and actions is a question to be debated and experimented with [3], as the different technologies used to produce different qualitative and quantitative results may indicate greater or lesser adherence to a CE vision model. The comparison with linear economy options is inevitable. "Take–use–dispose" or "take–make–dispose" is the linear economy strategy, traditionally used in rainwater and stormwater management as "take–dispose" [4,5]. Sustainable management, closely linked to technologies known as alternative, decentralized, and GIs, such

as wetlands, dry wells, green roofs, pervious pavements, retention ponds, tree trenches, and bioretention cells, is more in line with the circular model, as it takes care of the resource as close as possible to the source and seeks its exploitation. This implies reducing runoff and increasing infiltration, storage, and, often, recharging aquifers and a consequent improvement in the quality of runoff water during the urban processing of this resource, protecting it and allowing its reuse while providing it with greater value, as opposed to simple disposal (reuse principle). Increasing infiltration and reducing runoff also reduce the amount of stormwater directed to treatment plants, bringing chemical and energy savings, especially in the case of unitary or combined systems (which bring together wastewater and stormwater in a single pipe). In this way, energy and the corresponding emissions are reduced, as well as the number of chemicals needed for treatment (reduce principle), thus making the process more efficient and less expensive. For the combined sewer overflow (CSO), an increasingly frequent occurrence [6], the green infrastructure (GI) options, by reducing the volumes and peaks, also enable the reduction of CSO, contributing to a better quality of the environment [7] and reducing CSO emissions in receiving bodies [8].

Concerning the technologies used, wetlands are systems capable of transforming water quality by removing pollutants while enabling their storage and infiltration, bringing them to various quality stages so that they can be made available for reuse [9]. Dry wells have been used in locations with high precipitation volumes, and although they generate concerns regarding contamination from pollutants carried by runoff, it has been found through studies that this can be avoided when used in appropriate locations, considering the control of land use in their surroundings and proper maintenance [10]. Studies have also shown that the collection and storage of water from 10% of a region's rooftops can be translated into 1% less runoff in each studied region [11]. There are some technical obstacles to using rainwater and stormwater. Their use is not without risk, particularly because of the pollutants they can carry, although they can be reduced and treated. The demand for urban storage space and infiltration devices are also important issues, but largely rely on already-known solutions.

The main objective of this article is to examine the adoption of the CE concept, which originated from industrial ecology, built on the notion of resource utilization in a loop-closing way [12], and apply it to urban rainwater and stormwater management. With ideas and case studies, the advantages, barriers, and limitations of their application are illustrated and comments are made regarding their insertion in the sustainability concept and within the SDGs. The intended contribution of the article is to clarify the idea of urban rainwater and stormwater as a resource, still little used, but that should be treated as part of resource recovery. Resource recovery factories (RRFs) or just "water factories", thus considered wastewater treatment plants [13], are a little-explored part of the problem. Stormwater management involves not just wastewater treatment but also urban water supply optimization, energy and chemical savings, urban heat island (UHI) control, scarcity reduction, urban food, and, in short, increasing the citizens' well-being with cost reduction without necessarily worsening lifestyles and reducing sustainable development.

The objectives of this text are also to: (1) demonstrate by means of studies a trend in urban rainwater and stormwater use; (2) demonstrate that the CE of rainwater and stormwater is viable where there are conditions for a paradigm shift, from linear to circular, based on a mindset change by resource efficiency use; (3) demonstrate that CE may be in syntony with the SDGs; and (4) demonstrate that without the correct and aligned PIR frameworks it is difficult to implement CE. With these objectives in mind, we can contribute to the implementation of a stormwater CE, pointing out an efficient vision of the urban water cycle. This document is structured in six sections in addition to this brief Introduction. Section 2 deals with policy, institutions, and regulation, and Section 3 addresses CE of urban water. Section 4 presents the case studies and Section 5 discusses the main results. Finally, Section 6 draws the conclusions.

## 2. Policies, Institutions, and Regulation

CE requires necessarily a policy, institutional, and regulatory (PIR) framework capable of supporting CE initiatives. The PIR framework, however, is absent in most places because there are other kinds of barriers, some of them large, such as cognitive, normative, and regulatory aspects. Policies should pursue defined goals based on consensus among society's actors. The main goal of the CE is sustainable, efficient development. It should be carried out by maintaining the stock of natural resources, and to this end, policies that aim for the three Rs, reduce, reuse, and recycle, of all kinds of resources, including chemicals and energy, are needed to avoid their depletion and transformation into waste and emissions, disposed of in the natural environment.

In the CE context, the preservation of the environment, as a source of natural and economic wealth for future generations, involves the reduction of water, energy, and resource use, pollution and greenhouse gas (GHG) emission control, contribution to urban food, and thus the development of policies, institutions, regulations, and actions, creating governance focused on the CE principles. The CE Action Plan (CEAP), adopted in 2020 by the European Commission (EC), published on 11 March 2020, is part of the European Green Deal, the new agenda for sustainable European growth, having as one of its objectives the reduction of the pressure on natural resources and the creation of jobs. CEAP has a set of law and non-law measures that incentivize the adoption of CE and establishes a product policy framework that will make sustainable products, services, and business models usual. By this, the consumption patterns will be transformed so that no waste is produced in the first place [14]. The initiative encompasses the useful life of resources and products, seeking to make them circulate as long as possible within the economy, use them in a sustainable way, prevent the generation of waste, and enable the aggregation of greater value, consequently adding greater competitiveness. In terms of the European Union gross domestic product (EU-GDP), studies estimate that applying the CE model can produce 0.5% growth by 2030 and create about 700,000 jobs [15]. EU Regulation 2020/741 sets minimum requirements for the quality aspects of water reuse and this rule came into force on 26 June 2023, bringing concerns about the criteria, risks, and their standardization, which may pose a barrier to its implementation and funding of actions that consider it. The European Union (EU), for example, to improve the health and well-being of its citizens and to address climate change and environmental degradation, has built an agreement with targets of no greenhouse gas emissions by 2050, economic growth decoupled from resource use, and no people or places left behind. For this, an intermediate reduction target was established of 55% of GHG emissions by 2030, based on 1990 emission levels. One-third of the EUR 1.8 trillion investment in the Next Generation EU Recovery Plan has been earmarked for this, and the seven-year budget will fund the European Green Deal. Beyond the more general objective, however, what are needed are institutions capable of supporting actions between actors, as well as making explicit achievable goals with time horizons, ways, and mechanisms to measure and regulate them [16].

In China, the CE has been a policy option since the 2000s, capable of dealing with the high consumption of natural and environmental capital, so that the development may occur as quickly as possible. However, for its results to be verified, it was necessary to build several indicators and a legal framework that institutionally supported the actions. Thus, the Chinese central government introduced an institutional framework like the supporting legislation: in 2003, the "Cleaner Production Promotion Law", in 2005, the "Law on Pollution Prevention and Control of Solid Waste", and, in 2008, the "CE Promotion Law", which became the main frameworks to support the policy and regulation of actions related to the CE [2]. One of the central ideas that permeates the entire conceptual and legal framework is that it is possible, by means of the CE, to expand development with reduced impacts, i.e., decoupling growth from impact, which, traditionally, in the linear economy view, was believed not to be possible. When dealing with Chinese industrial production, we must consider that China, in practice, has become a major industrial player in the world, and this factor alone can already give an idea of the magnitude of the implementation of

CE in that country, given the large consumption of energy and water, in addition to all kinds of virgin materials [2] and their emissions.

## 3. CE of Water in Urban Environments

In general, but still in a simplified way, we can analyze the circularity of the processes that involve products, examining their degrees of circularity through indicators or indexes.

When it comes to the subject of urban water (rainwater, surface water supply, groundwater, stormwater, wastewater, and desalination water) and its processes, we can analyze them with a focus on circularity, covering the three Rs: reduce, reuse, and recycle. Concerning the linear economy, the water that enters the urban water cycle, as inputs to a process, may have an initial purpose to be used or not, but it will always have an end destination, which is its disposal as waste. In the CE, both surface and groundwater, as well as rainwater, can be used and recovered for new uses as often as their quality and treatment allow. Both rainwater and wastewater, traditionally considered unusable, have been understood, for some years and in some places, as capable of being used for various purposes, in direct non-potable and indirect potable reuse. Thus, the cooling of UHIs, amenities (lakes, etc.), irrigation of gardens, cleaning of public spaces, sanitary discharges, car washing, industrial uses, and urban agriculture are some of the applications. Circularity, however, depends on the process involved in its management and the application chosen, and, because of this, it is difficult to measure. The most appropriate and comparable methods have yet to be defined so that decision makers in each municipality can make the choices that best suit them.

An important issue is circularity in an integrated way between the various forms of urban water and the energy involved in processes. When we use water as a resource in the processes, according to the linear economy, its origin is of relative importance, but from the point of view of a CE, its origin and the process of obtaining, distributing, and using it are relevant, also involving the energy, emissions, pollution, and chemical consumption aspects of its treatment. From this point of view, the issue of losses is important, that is, the reduction, the first of the three Rs, is considered a priority concerning the other two, because there is no point in having high rates of reuse and recycling if we have, at the same time, a high consumption related to what is produced and a high rate of losses, or waste, in the origin and distribution process, e.g., leakages in connections and pipes. That is, if the processes have high inefficiency in the origination, distribution, and using of resources. Thus, talking about CE implies talking about process efficiency, that is, from the origin to the end. In the case of water distribution and the linear economy approach, acceptable water loss rates are between 8 and 10%, below which fault detection and removal are considered costlier than the benefits arising from correction. Moreover, achieving water losses between 6 and 8% requires the implementation of systems with high costs [17]. In Europe, in twenty-nine EurEau member countries, the mean values for losses in the two units most used by professionals are 23% and 2171 $m^3$/km/y (volume per unit pipe network) and include all non-revenue water which might include water used for institutions, maintenance, street cleaning, firefighting, and others [18]. In Brazil, the national average losses are 40.1% and can reach, in some regions, more than 70% [19].

In circular terms, these figures must be evaluated regarding stormwater utilization, as other aspects must be considered, such as technology, which plays a determining role in costs (e.g., cost of using current high-density polyethylene (HDPE) pipe networks instead of iron and steel, formerly more widely used), bringing new issues to the calculation of circularity. Determining what percentage of stormwater loss is acceptable is still an open question, not only regarding physical losses but also regarding the energy consumption for its use (kWh/$m^3$). Concerning the water supply and wastewater, energy costs are in the range of 5 to 30 percent of total operating costs and can reach 40 percent in developing countries [20]. Wastewater treatment involves four times more energy than water supply, mainly due to sludge management, which accounts for 50 percent of total operating costs and 40 percent of GHG emissions [21]. Sludge, however, allows for resource and

energy recovery [22]. As technologies that enable water reuse and recycling are energy and chemical intensive, the analysis must compute them and therefore not boil down to the physical quantities of water reused or recycled [23]. The impact arising from these intensive uses must be part of the analyses of the processes involved as well as their costs. Understanding this complex scenario has been supported by the life cycle assessment (LCA) tool, a technique that considers the lifetime of the product, or resource, and includes its losses throughout the process, as per International Organization for Standardization (ISO) documents, like ISO 14040-44: 2006 (LCA) and ISO 14045: 2012 Eco-efficiency [24]. Challenges regarding energy efficiencies involve governance, knowledge gaps, and barriers to financing.

From a global point of view, climate change produced using various processes has often been cited, and in this aspect, limiting greenhouse gas (GHG) emissions emerges as important for the sustainability of the planet. Wastewater treatment plants emit methane ($CH_4$) and nitrous oxide ($N_2O$) and use energy in pumps and aeration; water supply and stormwater are energy and material intensive. Emission levels from processes involved in urban water are around 14% of global allowable emissions. However, in Denmark, for example, they account for only 1% [25], which seems reasonable, but it is also necessary to consider water reuse and energy recovery in all processes of the urban water cycle.

In China, indicators have been determined at the micro (of individual companies), meso (of industrial parks), and macro (eco-cities or eco-regions) levels. Macro indicators are used for a general assessment of CE development at national and regional levels, serving for planning future CE development, and meso indicators serve to analyze circularity at the level of industrial parks [12]. Regarding water, at the meso level, these indicators are classified into four groups: resource extraction (total quantity); resource consumption (per unit of industrial value produced and water consumption per unit of product); resource use (water reuse rate); waste disposal and pollutant emission (total wastewater discharged and emissions produced). At the macro level, water extraction includes surface water sources, groundwater, recycled wastewater, rainwater, and desalinated water and has seven indicators: extraction per unit of GDP; extraction per industrial value added; consumption per productive sector; industrial water reuse rate; wastewater recycling rate; total industrial wastewater effluent; and irrigation water use coefficient. In several countries, such as the USA, Japan, and the Republic of Korea, indicators derived from the material flow analysis (MFA) method have been used, based on the flow of materials and energy, seeking mass balance in which inputs equal outputs, based on the laws of thermodynamics. This method, however, proves to be more suitable for analysis at the macro level, suggesting the need to create other tools for approaching the micro and meso levels, which China sought to do through the CE indicators, coupling the two methods [2].

In terms of global analysis, however, LCA alone is widely accepted for individual process analysis, including the water management sector. Despite making it possible to decide among the available technologies, it does not prove sufficient to make choices that meet planet boundaries (PBs) and even SDGs such as SDG6 (water and sanitation for all) and SDG11 (sustainable cities). To do so, that is, not to exceed the planet boundaries (PBs), we need to define local limits, or nationally determined contributions (NDCs) [25]. Studies estimate limits as, for example, 14% of total GHG emissions related to urban water management [26]. Thus, for the processes to be comparable across the various studies, a metric needs to be defined, and the allowable emissions per person receiving the services can be calculated based on PBs at the value of 522 kg $CO_2$ equivalent, which corresponds in global warming terms to 1 W/m² or 1.06 °C [27], which is a more restrictive value than the one that was set as a target by the Paris Agreement, i.e., 1.5 °C [26]. The development of appropriate metrics enables the evolution of an emissions market, which can be an additional element contributing to the reduction of emissions.

## 4. Case Studies

### 4.1. North America—USA (Florida)

In some places, such as in the city of Orlando (287,000 inhabitants) in Florida, USA, a study including simulations was carried out encompassing the food–energy–water–wastewater (FEWW) nexus, focusing on the CE model [28]. Various institutions, the rules of the game [29], and organizations are demanded to establish processes and relate them in the direction of quantifying circularity. A dynamic system model (SDM) was used to evaluate economic circularity by using the Systems Thinking, Experimental Learning Laboratory and Animation (STELLA) 10.0 software, as a model capable of demonstrating energy and material flows, considering FEWW, multiple layers, and the interconnections between sectors from the perspective of climate change and various policies, such as energy recovery and stormwater reuse.

Concerning water, the study considers, among the organizations and processes, two water reclamation facilities, Water Conserv II (WCII) and Eastern Water Reclamation Facility (EWRF). For energy, the electricity came from a landfill (Orange County Landfill (OCL)), a power plant of diversified origin (coal, natural gas, landfill gas, solar photovoltaic) (Curtis H. Stanton Energy Center (CSEC)), and several photovoltaic farms (PV farms). Regarding food and urban agriculture, the central figure is the East End Market Urban Farm (EEMUF). All processes interrelate with at least one of the others in terms of input and output flows, forming economic circularity in a closed loop. In the scenarios evaluated, those with the incorporation of stormwater reuse (wet retention ponds) and renewable energy production proved to be more resilient from the point of view of cost–benefit–risk tradeoff analysis when using multiple criteria for decision making. The incorporation of stormwater reuse (wet retention ponds) as an additional source of water has shown that it is possible to irrigate urban agriculture, increasing food resilience and improving the water use resilience index (WRI). It is noted that the estimated growth of precipitation in Florida according to the Intergovernmental Panel on Climate Change (IPCC) is 10 to 20%, depending on the emissions scenario. At the same time, the runoff predicted for the 2050s–2080s by the Environment Protection Agency (EPA) Stormwater Water Management Model (SWMM), the EPA-SWMM 5.1 model, is 80–118% higher for Florida urban coastal basins [30]. Stormwater reuse and reclaimed water have the cost–benefit–risk tradeoffs shown in Table 1.

**Table 1.** Cost–benefit–risk tradeoff examples for water and wastewater in Orlando urban FEWW nexus system simulation ([28], adapted).

| FEWW Sector | Cost | Benefit | Risk |
|---|---|---|---|
| Reclaimed water | High capital and O&M. Energy-intensive treatment process. Distribution system investment. | Increased water reuse. Public use quality control regulations. Water supply dependence reduction. Water resiliency and sustainability. Alternative irrigation source. | Distribution and use restricted by regulations. |
| Stormwater reuse | Initial capital costs. | Low or minimal capital and O&M cost. Quality and quantity control. Water supply reduction. Alternative irrigation source. Groundwater and aquifer recharge. | Stormwater transport of pollutants and sediments. Large area for subsurface storage. |

Increasing use of urban GI for stormwater management, together with urban emissions control policy, makes us also consider its ability to sequester and store $CO_2$ emissions, the main gas among GHGs, as part of the economic circularity of water. According to studies by Chen [31], in the 35 largest Chinese cities in 2010, urban green spaces accounted for 6.38% of the total area of these cities, or 53.7% of the green spaces in all 657 Chinese cities. In terms of carbon sequestration, the total estimated in the 35 cities was 18.7 million tons, averaging 21.34 t/ha. In 2010, the amount of carbon sequestered was 1.90 million tons with an average of 2.16 t/ha/year, equivalent to sequestering only 0.33% of fossil fuels, but with

the expected maturation and growth of vegetation, it is predicted that soon the sequestered amount could be substantially higher. Table 2 presents comparison values.

**Table 2.** Countries' urban greenspace carbon sequestration average values.

| Country | Carbon Sequesters (t/ha/Year) | Reference |
|---|---|---|
| Republic of Korea (Seoul) | 0.5–0.8 | [32] |
| Canada (Vancouver) | 0.9 | [33] |
| Singapore | 1.4 | [34] |
| China | 2.2 | [31] |
| United States (Florida) | 2.5 | [35] |

### 4.2. North America—Canada

The adoption of green systems should consider not only their results in terms of reduction of possible and probable floods, a measure of the hydraulic and physical efficiency of these systems, but also implementation, operation, maintenance, and decommissioning costs, in an LCA perspective, as an economic efficiency analysis. The issue is not, however, limited to these measures, as the capabilities for reducing and extracting pollutants (e.g., nitrogen and phosphorus) and, especially, the total balance of GHG emissions, throughout the lifetime, must be analyzed. The difficulty in implementing a CE, therefore, lies in the existence of data and the ability to determine the balances of energy and emissions and the circular flow of materials, considering the three Rs.

Concerning the circularity of stormwater, in addition to these barriers, there is the challenge brought by the unpredictability of future precipitation due to the lack of knowledge of the real future scenario of global warming, to which we will all be subjected, both in terms of temperatures (+1.5 °C, +2.0 °C, +2...?), the rising ocean levels, melting glaciers, and other effects. Most studies consider local conditions and partial effects of the implementation of various GIs, such as in the Province of Ontario, Canada [36]. Ontario has a policy of low-impact development (LID) with seven alternative types of GI (bioretention cells, downspout disconnection, dry wells, green roofs, porous pavements, rainfall harvesting devices, and soakaways pits) simulated and analyzed to enable the selection of the best alternative for the creation of a planning tool, from economic and regulatory points of view, considering an LCA and life cycle cost (LCC) approach: construction, operation, maintenance, decommissioning.

### 4.3. Europe—Italy (Bologna)

In the Italian city of Bologna, whose wastewater system is predominantly combined (stormwater and wastewater in one conveyor system), 728 km of pipes convey stormwater and wastewater, from an area of 5530 ha with $3.5 \times 10^5$ inhabitants, to the treatment plant via gravity and 14 pumping systems. In case of very heavy rainfall, so that the treatment system does not exceed its capacity, 122 overflow systems send the water directly into the rivers. Studies carried out in a subbasin of Bologna, the "Fossolo" catchment, with about 10,000 inhabitants and of 48 ha, simulated several possibilities of quantity (volumes, in $m^3$) and quality control of total suspended solids (TSSs) in Kg/ha. The placements of "end of pipe" (detention tanks) and "source control" (storage reservoirs for water reuse or rainwater-harvesting systems (RWHSs), permeable sidewalks, and green roofs) devices were analyzed, both with and without the placement of real-time control (RTC) systems. One of the assumptions considered was that storage reservoirs, to be economically sustainable, should be responsible for 60% of the volumes of non-potable water to be consumed in the study area. For detention reservoirs, the volumes were 10 $m^3$/ha and 50 $m^3$/ha of impermeable area upstream, respectively, for each of the two modeled scenarios and with an outflow of 3 L/s ha for both scenarios. The results showed that the placing of detention tanks with 10 $m^3$ capacity and real-time control (RTC) would practically eliminate CSO and that the best alternative for reducing volumes and TSSs are storage tanks with RTC [37]. Rainwater storage, among all the technologies analyzed, appears to be

the most efficient in terms of quantity, which can save water supply (reduction and reuse principles), and quality, which can save energy and material costs with treatment (efficient use). These are requirements for circular economy.

### 4.4. Europe—Germany (Munich)

Using a central district of Munich, Maxvorstadt, as a hypothetical study territory based on the water–energy–food nexus, various aspects of resource utilization were addressed in comparisons between the currently existing, centralized system and the possibility of opting for decentralized systems [38]. Thus, aspects concerning forms of management and recovery of rainwater and wastewater in a circular economy perspective were analyzed, especially in terms of costs, starting from the assumption that these waters are valuable resources that must be utilized.

Munich's sewage system is of the combined sewerage type, e.g., it carries wastewater and stormwater together and is between 50 and 100 years old. The study area is 90% sealed and densely inhabited with 46,960 people (12,000 people/km$^2$), suffering the effect of urban heat islands (UHIs). The water consumption is 6 million liters per day, of which about 27% are spent in sanitary flushing, generating a sanitary sewage volume of about 4 million m$^3$ per year, or 82 m$^3$/capita per year, without considering rainwater runoff. The study involved the recovery and reuse of water, the potential for energy recovery, and the recovery of nutrients for use in urban agriculture. In 2015, it was estimated that the system received for treatment 19.6 million m$^3$ of stormwater, or about 12% of the total of 165.5 million m$^3$, in treatment plants. The municipality charges 1.3 €/m$^2$ of impermeable area and 1.56 €/m$^3$ of water disposed in the sewage network and the per capita cost of maintaining the system is 14.94 €/capita per year, with about 50% being for maintenance and expansion of the network. The estimated cost for a decentralized system is 6.2 €/capita/year, or 0.95 €/m$^3$ of operational cost, and the cost of decentralized treatment was estimated at 60% of the cost of centralized treatment but this may be inaccurate. The conclusion of this hypothetical study is that the savings from the use of rainwater would be EUR 1.3 million per year (principles of reduction and reuse), adding EUR 2.8 million per year from the decoupling of wastewater treatment (recycle principle) and a further EUR 0.854 million per year for the energy savings (principle of reduction), totalling about EUR 5 million per year, taking two years to amortize the investments necessary to change the centralized system to decentralized, since this would have a cost of about EUR 10.4 million. The change to a decentralized system, however, would not mean the deactivation of the centralized system but rather its possible use as a preventive system against future flooding, due to predicted increases in precipitation due to climate change. On the other hand, the need for space for rainwater storage reservoirs is a strong economic constraint to the adoption of rainwater as a resource due to the high density of buildings. The provision of spaces must be in line with urban planning. The costs presented here do not consider the cost of storage spaces. It must be realized, however, that the costs cannot be extrapolated to other places, because of the various project scales and different amortization times, as well as the natural variation of cost constraints from one region to another.

### 4.5. European Mediterranean—Spain (Alicante)

In the direction of the CE, because of the effects of climate change, with the production of changes in rainfall patterns, increased flooding, and urban water shortage due to growing demand, the city of Alicante, southern Spain, a European Mediterranean area, adopted a policy of rainwater storage. To this end, it sought to build adequate water management infrastructure, making solutions for flooding and the growing demand compatible. The result was the reduction of demand on the water supply (reduce principle) and, simultaneously, adaptation to climate change with the use of rainwater (reuse principle) and the reduction of floods with consequent savings in resources [4]. The rainwater storage policy seeks to address two climate-related challenges: changes in rainfall patterns, less frequent

and more intense, and the consequent and simultaneous issues of water scarcity. For this, besides the two existing rainwater reservoirs (100 m$^3$), the construction of seven more is planned, reaching a storage capacity of 500 m$^3$. This total is equivalent to that of the city of Barcelona. In addition to the reservoirs, however, networks are needed to connect the treatment plants and connect them to the distribution networks that allow their use. La Marjal Floodplain Park, opened in 2015 as a reservoir with 45,000 m$^3$ capacity, is 4 km from the Monte Orgegia water treatment plant, the destination of the captured stormwaters. The storm drainage network system of Alicante is 113,000 km (17% of the total drainage network) and is a separate system from the wastewater drainage system.

The policy is accompanied by the institutional framework, present in existing legislation in several cities, such as Barcelona, Málaga, Alicante, Reus, Calviá, and different municipalities in the Basque Country concerned with reducing the effects of climate change. The regulatory framework is materialized by the Water Framework Directive 2020/60/EC. In Alicante, the infrastructures are also part of the Master Plan for the Reuse of Treated Water whose objectives include the replacement of drinking water for street cleaning and watering of green parks. The organization that approves the plan and supports the actions is the city's water supply company (Aguas Municipalizadas de Alicante, Empresa Mixta (AMAEM)), a water company owned in equal parts by the city council and a private company (Hidraqua, Gestión Integral de Aguas de Levante S.A.), a subsidiary of Aquadom (Suez Environment), managing the hydrological cycle. Also, due to popular public decisions in Spain, sustainable urban drainage systems (SUDs), low-impact development (LID), sponge cities, or green infrastructures, management stormwater strategies and technologies whose objectives are to mimic the pre-urban hydrologic and hydraulic conditions, are implemented. Rainwater harvesting (RWH) is one of these technologies. Although rainwater is low in cost at its origin, there are costs involved in infrastructure maintenance and treatment. For Alicante, [5] estimated 0.32 €/m$^3$ compared to 1.69 €/m$^3$ for conventional treated water supply. When compared to desalination costs, estimated at 0.46 €/m$^3$ for this region, it appears that it is more competitive to use stormwater.

*4.6. Asia—China*

In 1998, the CE concept was first proposed in China, but the change in focus from waste recycling to process improvement and efficiency with new technologies became, from the 2000s on, the center of the industrial reform brought by CE as an accepted state policy of the Chinese central government, through the National Development and Reform Commission (NDRC), and not only as an environmental policy. CE became one of the country's development strategies for the 21st century. The institutional support by laws starts with the Cleaner Production Law, in 2003, the amended Law on Pollution Prevention and Control of Solid Waste, in 2005, and in 2008/9, the Circular Economy Promotion Law of the People's Republic of China. More recently, the policy named Sponge Cities Initiative became the main solution for floods and the management of urban waters. Beijing, Shanghai, Dalian, and Tianjin are examples of cities included in the Sponge Cities Initiative [39].

In 2015, Wuhan, a city with an estimated population of 10 million by 2035, was declared one of the first sixteen sponge cities (first stage of a total of thirty cities, in two stages) to receive attention and funding for the implementation of sustainable stormwater alternatives. With a total of two hundred and twenty-eight projects in the two pilot districts of Qingshan and Sixin, more than 38.5 sq km of the city have so far received CNY 11 billion [40]. Wuhan is a city situated where the Yangtze and Han rivers merge and it has a relatively high level of groundwater, a characteristic that makes it difficult for the stormwater to go to the rivers during intense rainwater events. The policy of the Sponge Cities Initiative includes a pilot project, until 2030, when 80% of the areas must have urban lands with sponge characteristics and be able to retain 70% of the stormwater. The projects can have a public–private partnership which, in the case of Nanganqu Park, in Wuhan,

includes 20% government subsidies and 80% from the private sector, in this case, the iron and steel company that built the affected residential areas for workers in the 1970s and 1980s.

Already at that time, there was the perception that the linear model of economic development was unsustainable from the point of view of resource productivity and eco-efficiency, and that the strategy defined, using a CE model policy, promised to be the appropriate response, despite not yet having the necessary policies, institutions, and regulation requirements for its implementation [13]. Thus, in this way, in China, they thought that it would be possible to achieve sustainable development, i.e., without the exaggerated consumption of its resources.

## 5. Discussion

A green economy policy alone does not mean a CE orientation, for example, a country may have a green policy for using renewable energy sources (solar, wind, hydroelectric, biomass, and geothermal) without having a CE policy. CE, by reducing the use and extending the permanence of resources in the economy, brings, as a result, the opportunity to create additional value, producing efficiency gains by reusing and recycling them [41]. The CE can be considered as part of sustainability, even though these are often two concepts that can be confused due to some similarities, as the CE, in general, contributes to sustainability. However, it does not focus on social dimensions, unlike sustainability, which is based on the tripod of economic performance, social inclusivity, and environmental resilience [42]. On the other hand, from an institutional point of view, while sustainability is a broad concept, dependent on the alignment between actors, that adapts to different contexts, CE, when compared to linear economy, has a greater responsibility and emphasis on governments, regulators, and companies with a focus on economic and environmental benefits, prioritizing financial advantages for companies and decreased resource consumption and pollution for the environment [42]. The CE requires the alignment of actors around policies, institutions, and regulations that support it, and thus its application, more than allowing the balance of resources with efficiency, must exceed the limits of the principles of economics and thermodynamics and take into account the environmental and social aspects, present in the approach to sustainability of the planet, of which the CE is part or is intended to be part [43].

In terms of the application of the new ICTs available, they can contribute decisively, since they make possible the distributed measurement of parameters that were previously only commonly measurable in centralized facilities such as wastewater treatment plants (WWTPs) and water treatment plants (WTPs) and at some points in the distribution networks, due particularly to the costs and difficulties of measurement and data transmission. In this respect, the measurement of stormwater parameters at various distributed network points in cities makes it possible to quantify the circularity of the water [44]. These technologies can increase the interest and participation of the population in urban water management, for example, by allowing images to be transmitted via smartphones in real time, showing leaks, network losses, defects, or even flood events and their consequences [45]. It is a kind of extension of the idea of citizen science, defined by the European Commission as "general public engagement in scientific research activities where citizens actively contribute to science either with their intellectual effort, or surrounding knowledge, or their tools and resources" and involving various forms of public participation [46].

Regarding climate change, emissions reduction, and adverse effects such as intense rainfall and water shortages, the last International Conference of the Parties (COP-27), held in Egypt in 2022, reiterated several approaches to combat climate change and the resulting material and financial losses and emphasized financial and technology transfer aspects. As for the financial aspects, COP-27 pointed out the need for USD 4 trillion per year to be invested in clean technologies by 2030, to extinguish emissions by 2050. The transformation to a low-carbon economy will require between USD 4–6 trillion per year [47]. Developing countries require approximately USD 5.8–5.9 trillion by 2030, pointing to the need for support for these countries by more developed ones. The uneven ability to raise funds,

transfer technology, and build capacity for climate change mitigation indicates the need for reform in the practices of multilateral organizations and banks to streamline risk analysis and funding procedures. In Central Asian countries (CACs) (Kazakhstan, Turkmenistan, Tajikistan, Kyrgyzstan, and Uzbekistan), for example, there are opportunities for implementation of the CE because they have legislation oriented towards the green economy and written commitments to sustainability. There is also the interest from international organizations, which can mean funding, and schools and universities that adopt environmental education, but among the barriers to the implementation of the CE, there is, according to Tleuken, an absence in the legislation of a clear path forward towards goals and a lack of circularity policies, which means an absence of regulatory instruments, without which CE does not exist [48]. In summary, like the concept of sustainable development, of which CE can be understood as an integral part, CE is not a panacea to be applied in all situations, as the examples presented here demonstrate, and it requires a framework of policies, laws, regulations clearly understood by society for its application.

## 6. Conclusions

The adoption of a CE policy requires an underlying change in mentality regarding the use of resources, in addition to aligned PIR frameworks that can provide incentives for CE application and an interdisciplinary vision that involves the water–stormwater–wastewater–food–energy nexus in urban environments. Furthermore, it is necessary to consider uniform metrics for circularity, including the PB approach, considering regional and local issues, that allow for its compatibility with the planet's limits. This vision, however, is not yet present in most cities. How and where to start this transition are open questions. Some initial issues were identified, such as the challenge of circularity calculation. Policies put in place, such as in China and the EU, are a good start. Ultimately, it is a matter of improving the efficiency of processes that involve the use of resources, of which stormwater is one but little recognized and understood as such. Stormwater, however, should be looked at from the perspective of its integration with the other urban waters and all of them in a broader perspective, i.e., of the urban water cycle with compatibility between all resources and processes. Inside this context are energy, chemical products, waste, and food, seeking their optimization, to combat water scarcity and UHIs and produce a reduction of GHG emissions, to be within the existing total PBs. In short, adopting CE, including stormwater, may shorten the path to intergenerational sustainability. Putting urban stormwater in the context of water use efficiency within the urban water cycle is an important issue in the CE. In each specific location, the degree of efficiency or waste of water is different, a fact that conditions the possibility of the existence of CE of water and stormwater. There is no sense in talking about reuse and recycling when there is no reduction of water use at its origin or waste in its distribution and use, causing enormous losses. Thus, initial efficiency, considered by calculating losses in extraction, distribution, and use, is a basic assumption for considering stormwater in the context of CE. The measurement of consumption, in view of basic human needs and degree of development, can signal inefficiency in the processes. PIRs can contribute greatly to determining the efficiency needed to move towards CE [49]. Countries with primary inefficiencies such as no water supply, or where part of the population is outside water systems or where water losses are large or barely measurable, are far from the CE concept. The inclusion of rainwater and stormwater in the water cycle within the idea of CE may mean that significant parts of the world's population may be included in water services. More than that, the inclusion of urban rainwater and stormwater may be a way to solve scarcity and help achieve sustainability goals.

It is recommended that future research focuses on the limitations found regarding the interrelationship between environmental and social sustainability aspects and CE, trying to show how this link is found, especially through case studies and experiments. Another aspect that is not very present concerns the understanding and participation of society, as the main actor, interested in urban sustainability and, consequently, in policies related to CE.

Future research should explore what the wishes of the population are in the implementation of CE.

**Author Contributions:** Conceptualization, C.N. and R.M.; methodology, C.N. and R.M.; validation, C.N. and R.M.; formal analysis, C.N. and R.M.; investigation, C.N.; resources, C.N. and R.M.; data curation, C.N. and R.M.; writing—original draft preparation, C.N.; writing—review and editing, C.N. and R.M.; visualization, C.N. and R.M.; supervision, R.M.; project administration, R.M. All authors have read and agreed to the published version of the manuscript.

**Funding:** This research received no external funding.

**Institutional Review Board Statement:** Not applicable.

**Informed Consent Statement:** Not applicable.

**Data Availability Statement:** No new data were created.

**Conflicts of Interest:** The authors declare no conflict of interest.

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
