# Peer review of "Are Rainwater and Stormwater Part of the Urban CE Efficiency?"

_sustainability, doi:10.3390/su151411168_

Round 1

Reviewer 1 Report

In the manuscript (sustainability-2485026), the authors investigated the adoption of the CE concept in urban stormwater management, which offered an interesting perspective. However, there are some defects in the current manuscript. Some suggestions may offer a reference for the authors.

1. The authors mentioned two words, rainwater and stormwater, in the title of the manuscript. What’s the difference? The authors may add more discussion on the two words as they are mentioned in the title but not discussed or explained in the text.

2. The whole structure of the manuscript should be reorganized. The abstract did not fully reflect the intention, core issue and result, as well as the value of the study. The introduction seemed scattered, which should be rewritten to 3-5 paragraphs. Other sections, including sections 2-5 and the conclusions, have similar problems. The author should read the instructions for authors and other published papers in the journal of Sustainability or other journals to improve the organization of the manuscript.

3. The cost and price of 3Rs of rainwater are core issues and limitations that affect the application, which were not mentioned and analyzed in the manuscript. I would recommend the authors add some discussion and analysis if possible.

4. The manuscript is closer to a review, while as a review, the depth is not enough and the number of studies reviewed and discussed in the manuscript is of great shortage.

Extensive editing of English language required

Reviewer 2 Report

Review of the manuscript entitled Is rainwater and stormwater part of the urban CE efficiency?

The water management in urban areas is a challenge for decision makers. Those areas, determine specific water circulation and as a result there are many hydrological problems related to excess and shortage of water. The authors in the article discussed on the aspects related to use of the rainwater in context of the circular economy paradigms. In rewiever’s opinion the topic is important, especially in urban areas, where water excess may be a reason of e.g. urban floods.

The authors used the meta-analysis approach and, on the basis of the literature discussed main elements which may contribute to the changes in the policy of urban rainfall water management process. In the second part the authors presents the examples of good practices described in the literature. The final conclusions, emphasised the main aspects of the topics discussed in the article.

 The meta-data analysis performed in this study are very interested and the article have potential to be cited and used in as a part of the discussion in this topic. The article is well written. The goals are clearly defined. The discussion sections explain the most important aspects in relation the goals.  I recommend publication of the manuscript in this form.

Reviewer 3 Report

The authors undertook an important task concerning the analysis of the need for water retention in urban areas. In the era of changing climate trends, solutions allowing for the retention of excess water and then its use during drought. Of course, the concept is not new, but adapting the system to modern needs and taking into account climate balancing is very important nowadays.

The work is very interesting, clearly written. The authors presented the problem and the purpose of the research well. The analytical part is not objectionable. It is a pity that the authors did not attach figures, diagrams, photos. Visualization would be interesting. Since the title of the article is in the form of a question, in my opinion the best justification in the summary would be a SWOT analysis. This is not a recommendation, but a proposal.

In my opinion, the article is very interesting and fits into the trend of pro-ecological solutions in the design of sustainable cities. It is worth publishing.

Round 2

Reviewer 1 Report

Dear authors,

Thanks for your effort in the improvement of the manuscript. I'm very glad to see many revisions corresponding to the reviewers' comments and suggestions. 

I still have one small concern about the paragraph structure of the manuscript. I hope the authors may adjust or reorganize the paragraphs since the current paragraphing seems scattered. 

Other concerns in the first round have been well addressed. 

NA.
